# LEARNING TO SUMMARIZE USER INFORMATION FOR PERSONALIZED REINFORCEMENT LEARNING FROM HUMAN FEEDBACK

**Hyunji Nam**
Stanford University
`{hjnam}@stanford.edu`

**Yanming Wan**
University of Washington
`{ymwan}@cs.washington.edu`

**Mickel Liu**
University of Washington
`{mickel7}@cs.washington.edu`

**Peter F. Ahnn**
DoorDash
`{peter.ahnn}@doordash.com`

**Jianxun Lian**
Microsoft Research Asia
`{jianxun.lian}@microsoft.com`

**Natasha Jaques**
University of Washington
`{nj}@cs.washington.edu`

## ABSTRACT

As everyday use cases of large language model (LLM) AI assistants have expanded, it is becoming increasingly important to personalize responses to align to different users' preferences and goals. While reinforcement learning from human feedback (RLHF) is effective at improving LLMs to be generally more helpful and fluent, it does not account for variability across users, as it models the entire user population with a single reward model, meaning it assumes that everyone's preferences are the same. We present a novel framework, **P**reference **L**earning **U**sing **S**ummarization **(PLUS)**, that uses reinforcement learning (RL) to learn to produce text-based summaries of each user's preferences, characteristics, and past conversations. These summaries condition the reward model, enabling it to make personalized predictions about the types of responses valued by each user. Both the user-summarization model and reward model are trained simultaneously, creating an online co-adaptation loop. We show that in contrast to the standard Bradley–Terry model, summaries produced by PLUS capture diverse aspects of user preferences, achieving a 11–77% improvement in reward model accuracy. Key strengths of PLUS are: (1) robust performance with new users and conversation topics, achieving a 25% improvement over the best personalized reward model technique used for RLHF; (2) personalization with strong proprietary models like GPT-4 without further fine-tuning (e.g., PLUS-summary-conditioned responses achieved a 72% win rate compared to 28% for default GPT-4o); (3) learning from flexible user contexts beyond preference labels, and (4) interpretable representation of users, enabling greater transparency and user control in pluralistic LLM alignment.

## 1 INTRODUCTION

Large language models (LLMs) have become a daily part of life for over a billion people, with Gemini reaching at least 450 million monthly users by July (Lee, 2025) and ChatGPT's daily users approaching a billion (Curry, 2025). Users interact with LLMs in various ways, from asking for domain-specific help to engaging in open-ended storytelling and content creation(Zheng et al., 2024). Reinforcement Learning from Human Feedback (RLHF (Ouyang et al., 2022)) is a critical component of modern LLMs deployed in production systems, because it ensures that they are not only functional (i.e., usefully follow user instructions) but also aligned with constitutional values such as safety and helpfulness (Bai et al., 2022). The standard Bradley-Terry-Luce (BTL) reward model (Bradley & Terry, 1952) used in existing RLHF algorithms learns a reward function based

on majority preferences: If two users have opposite preferences about the response to a particular prompt, and no further context is provided, the BTL model cannot learn to disambiguate between different users' preferences.

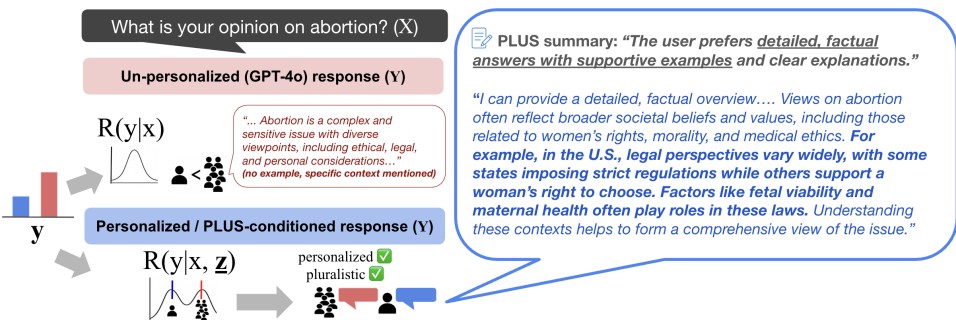

Figure 1: *Caution: This content may reflect particular beliefs.* While standard RLHF techniques fail to capture user variability, PLUS trains both a summarizer and reward model in an online co-adaptive framework to learn summaries $z$ useful for predicting diverse preferences. Italicized texts are actual outputs by GPT-4o and PLUS showing the effects of summaries on personalization.

However, real users have diverse, potentially conflicting, preferences (Kirk et al., 2024), and current methods are unable to accommodate these differences. In fact, when the standard BTL model is applied to data with conflicting preferences, it will either ignore the preferences of minority groups (Siththaranjan et al., 2024), or learn a reward model that is inaccurate for *all* groups of users (Poddar et al., 2024). This has led to recent calls to develop techniques for *pluralistic alignment* Sorensen et al. (2024); Li et al. (2024); Siththaranjan et al. (2024), to enable LLMs to be personalized to heterogeneous user preferences. Given the widespread use of LLMs and the diverse populations of users they serve, being able to model and adhere to more than one set of preferences is critical for making effective, accurate assistants that are capable of personalizing to each individual's unique needs. One way to achieve pluralistic alignment is by developing a reward model conditioned on user variability, rather than relying on a single BTL model (Siththaranjan et al., 2024; Zhao et al., 2024; Poddar et al., 2024) (e.g., (Poddar et al., 2024) learns a latent user embedding from the past preference labels). However, compressing rich text into a vector can lead to performance loss (Cideron et al., 2022). As yet, we are missing a highly performant, *public* RLHF technique capable of modeling the preferences of diverse users.

A more natural approach which retains the strengths of LLMs to reason with text, is to learn text-based user summaries to act as the latent user variable. We propose **Preference Learning Using Summaries (PLUS)** as a general RLHF framework that *learns* to summarize important information about the user using reinforcement learning (RL) fine-tuning and conditions the reward model on these user summaries. PLUS consists of two co-dependent training phases: (1) a summarizer trained with Proximal Policy Optimization (PPO) (Schulman et al., 2017), and (2) a summary-conditioned reward model, which provides the reward signal for training the summarizer and makes predictions about user preferences.

We carefully compare our approach to several alternative implementations. First, we check whether we can simply prompt an LLM to generate user summaries without fine-tuning the summarizer. Our experiments show that automatic summaries (even from GPT-4) often miss key details necessary to capture user preferences. Another alternative is to condition the reward model on the user's entire conversation history, without summarizing it. However, both approaches fall short compared to concise, summary-conditioned reward models when conversation topics change. An additional benefit of concise summaries compared to existing techniques is that the summaries are easier to review and edit, improving transparency and trust in LLM assistants. We show that such benefits extend to complex real-world datasets, like PRISM (Kirk et al., 2024), where diverse user preferences can be expressed through flexible textual summaries and used to enable personalization of strong, proprietary models without additional fine-tuning. In summary, our main contributions are:

**1. Novel pluralistic alignment algorithm, PLUS (Preference Learning Using Summarization)** that uses RL to jointly learn a summarizer and reward model. PLUS improves accuracy, interpretability, and transferability across new conversations and users.

**2. Empirical advantages of PLUS on benchmarks from prior state-of-the-art work** (Poddar et al., 2024): Pets and UltraFeedback (Cui et al., 2024). PLUS improves reward model accuracy by 11-77% compared to BTL, is robust to topic shifts, and can handle unstructured user contexts.

**3. Scaling to a dataset in the wild**. We are one of the first to report reward modeling results with PRISM (Kirk et al., 2024), a pluralistic dataset of 1,500 users across 75 countries and 20 LLMs.

**4. Enabling personalization of strong, proprietary models via PLUS-generated summaries.** PLUS can generalize to unseen users of the PRISM dataset and the learned summaries can be used to personalize responses of strong proprietary models without additional finetuning (PLUS summary-conditioned responses achieve a 72% win rate over the default GPT-4o responses).

## 2   RELATED WORK

**Reinforcement Learning from Human Feedback (RLHF).** Our work builds on reinforcement learning from human feedback (RLHF) (Ouyang et al., 2022), which models binary human preferences using the Bradley-Terry-Luce (BTL) model (Bradley & Terry, 1952). The BTL model assumes the user provides preference ratings in terms of which of two responses they like better; it then learns a reward model (RM) that estimates the magnitude of the user's preferences using this data. This is then used to train an LLM policy to optimize responses based on the reward signal. RLHF has been effective in refining language models' responses to better align with user-preferred attributes, like helpfulness and truthfulness (Bai et al., 2022). However, existing RLHF works (Ziegler et al., 2020; Bai et al., 2022; Ouyang et al., 2022; Siththaranjan et al., 2024; Rafailov et al., 2024) have assumed a single reward model to capture all user data, which may be misaligned with heterogeneous users.

**Personalized & Pluralistic RLHF.** Increasingly, more works (Sorensen et al., 2023; Kirk et al., 2023; 2024; Zhao et al., 2024; Poddar et al., 2024; Li et al., 2024; Chen et al., 2025; Liu et al., 2025) have pointed out the limitations of existing BTL reward models in accommodating diverse underlying preferences, which may conflict with each other in terms of what individual users consider ideal responses. These differences may surface along various dimensions, from writing style to value alignment. Despite algorithmic advances in pluralistic alignment, large-scale human datasets are still limited. PRISM (Kirk et al., 2024) makes an effort to provide a heterogeneous user dataset consisting of 1,500 participants from 75 countries interacting with 21 different LLMs (contributing around 9,000 unique conversations). However, so far we are not aware of work that has successfully published reward modeling results on PRISM, pointing to the difficulty of modeling diverse and heterogeneous preferences with sparse data.

Broadly, works on pluralistic RLHF have focused on either of two main aspects: (1) personalizing reward models (Siththaranjan et al., 2024; Poddar et al., 2024; Zhao et al., 2024; Liu et al., 2025; Chen et al., 2025) and (2) personalizing LLM responses with a user model (Li et al., 2024). Reward (user) modeling serves as the backbone of many existing algorithms for LLM alignment, since reward model accuracy has direct implications for downstream tasks, such as RLHF finetuning and best-of-N sampling (Shen et al., 2023; Gao et al., 2022; Meng et al., 2024). Notably, Poddar et al. (2024) introduces Variational Preference Learning (VPL) which uses a variational autoencoder to learn user-specific embeddings and conditions the reward model on these latent variables. Siththaranjan et al. (2024); Zhao et al. (2024); Chen et al. (2025) provide algorithms for accommodating diverse user groups through methods such as Distributional Preference Learning (DPL), dispersion index, and shared LoRA adaptation. However, the key difference between prior works and ours is that these methods represent users as embedding vectors or (linear) weights, which reduces the power and interpretability of the underlying personalization mechanism. In contrast, we leverage the text-generation capabilities of LLMs to represent diverse user preferences through summaries.

**LLM finetuning for summarization.** Finetuning LLMs for improved and domain-specific summarization has long been of interest (Stiennon et al., 2022; Ziegler et al., 2020; Zhang et al., 2023; Van Veen et al., 2023; Song et al., 2025; Wu et al., 2025). Most closely related to our work is by Wu et al. (2025) which demonstrate that learning user summaries can be effective for personalizing recommendation systems, when you assume access to a fixed user classifier. In contrast, we focus on the online co-adaptive loop of training both the summarizer and reward model, which serves as the key for enabling the reward model to make use of the learned summaries for capturing nuanced user preferences.

## 3 PRELIMINARIES

**Bradley-Terry-Luce reward model.** Our work focuses on the reward modeling stage of the standard RLHF pipeline (Ouyang et al., 2022). Given a dataset of annotated preferences $\mathcal{D} = \{(s_A^i, s_B^i, \mathbb{1}\{s_A^i \succ s_B^i\})\}_{i=1}^N$, where $s_A = [x, y_A]$ and $y_B = [x, y_B]$ are two responses $y_A, y_B$ over the same prompt $x$, the standard Bradley-Terry-Luce (BTL) model (Bradley & Terry, 1952) assumes an underlying reward function $r(s)$. This reward function determines the likelihood of the preference label $\mathbb{1}\{s_A \succ s_B\}$ as:

$$p\big(\mathbb{1}\{s_A \succ s_B\} = 1 \mid s_A, s_B\big) = \frac{\exp r(s_A)}{\exp r(s_A) + \exp r(s_B)} = \sigma\big(r(s_A) - r(s_B)\big), \qquad (1)$$

where $\sigma(x) = 1/(1 + e^{-x})$ is the sigmoid function. The reward modeling stage typically involves learning a parameterized reward function $r_\phi(s)$ by maximizing the likelihood of Eq. 1 over the observed response pairs and their preference labels. The BTL model assumes all users share the same reward function, and thus fails to capture conflicting preferences when two users prefer different responses to the same prompt $x$.

**User-conditioned reward model.** Existing work in personalized RLHF (Li et al., 2024; Poddar et al., 2024) has demonstrated the effectiveness of extending the standard BTL reward model into a user-conditioned reward model:

$$p\big(\mathbb{1}\{s_A \succ s_B|z\} = 1 \mid s_A, s_B, z\big) = \frac{\exp r(s_A|z)}{\exp r(s_A|z) + \exp r(s_B|z)} = \sigma\big(r(s_A|z) - r(s_B|z)\big) \quad (2)$$

where $z$ is an identifier (e.g., a user ID) that distinguishes different users or user groups. Eq. 2 is better suited for capturing the population's heterogeneous preference distribution that may be multimodal, and the target is to learn a parameterized user-conditioned reward function $r_\phi(s|z)$. This gives the reward model the expressivity to capture different user types, which yields more accurate predictions when user preferences are diverse and even conflicting.

However, in most real-world scenarios, datasets do not contain user identifiers $z$. This is partly because such identifiers may be unavailable due to privacy issue, and partly because conditioning only on users observed in the dataset makes it difficult to generalize to new users. Therefore, we can instead use context $c$, such as past conversations, to represent the current user in each data point[1]. This leads to a dataset of user-conditioned annotated preferences $\mathcal{D}' = \{(s_A^i, s_B^i, c^i, z^i, \mathbb{1}\{s_A^i \succ s_B^i|z^i\})\}_{i=1}^N$ where $z^i$ are unknown. For notational convenience, we assume $s_A^i$ is always the preferred one by user $z^i$, and the dataset can be simplified to $\mathcal{D}' = \{(s_A^i, s_B^i, c^i)\}_{i=1}^N$.

## 4 PREFERENCE LEARNING USING SUMMARIZATION: PLUS

While prior works (Li et al., 2024; Poddar et al., 2024) formulate the latent user representation $z$ as an embedding vector, we propose learning $z$ **in natural language** as a more natural way to leverage the advanced text-based reasoning and generation capabilities of LLMs. To be more specific, we propose to use a **summarizer** $\pi_\theta$ to generate a concise representation in natural language based on the context $c$ of each user, $z \sim \pi_\theta(\cdot|c)$. Then, the parametrized **reward model** $r_\phi(\cdot|z)$ will compute the reward conditioned on this learned summary $z$.

Rather than manually defining any ground truth criterion for useful user summaries and pre-training a summarizer in advance, we argue that the quality of a summary should be measured by its ability to enable the downstream reward model to accurately predict user preferences. Therefore, the summarizer and the reward model should be optimized under a joint objective, namely minimizing the negative log-likelihood loss based on Eq. 2 across the user-conditioned dataset $D'$:

$$\mathcal{L}_{\text{NLL}}(\phi, \theta) = -\hat{\mathbb{E}}_{(s_A, s_B, c) \sim \mathcal{D}', z \sim \pi_\theta(\cdot|c)}\big[\log \sigma\big(r_\phi(s_A|z) - r_\phi(s_B|z)\big)\big] \qquad (3)$$

To handle the co-dependency in Eq. 3, we design a training process that continues to alternate between training one model while keeping the other fixed. By allowing the summarizer and the reward

---

[1] Poddar et al. (2024) requires that the context consist of a user's past chosen and rejected responses to different prompts $c = \{(s_A^j, s_B^j)\}_{j=1}^m$. In contrast, PLUS can incorporate more flexible inputs like past conversations without explicit preference labels and system strings.

model to co-adapt, the summarizer learns to extract information that is most useful for improving the predictive accuracy of the reward model, while the reward model is trained on updated summaries to more accurately capture user preferences. This process is computationally efficient, since the generated summaries always target the most recent iteration of the reward model.

**Training the reward model.** When training the reward model, the summarizer weights are fixed, and the reward model is trained to minimize the negative log-likelihood loss (Eq. 3) using pairs of chosen and rejected responses, conditioned on user summaries sampled from the summarizer policy. Then, we repeat the step of training the summarizer using the newly obtained reward model weights.

Figure 2: Eq. 3 naturally leads to the online co-adaptation of the summarizer and the reward model.

**Training the summarizer.** Unlike the reward model, a summarizer's optimal weights cannot be learned directly through supervised learning, because whether a summary is effective for predicting preferences can only be assessed once the full summary has been sampled. Instead, RL can optimize the summarizer at the token level by assigning credit for the full-trajectory return across long token sequences $(z_1, z_2, ..., z_{t \le H})$, as demonstrated by prior successes in using RL to improve summarization (Stiennon et al., 2022; Ziegler et al., 2020). Thus, we formulate generating the summary into an RL post-training problem, with the objective in Eq. 4. We start with a default LLM as a reference policy $\pi_\theta$, sample summaries (trajectories) from the current $\pi_\theta$, estimate their returns (as negative losses of the reward model), and update the weights $\theta$ in a direction that maximizes estimated returns.

$$J(\pi_\theta) = \hat{\mathbb{E}}_{\substack{z \sim \pi_\theta(\cdot|c) \\ (s_A, s_B, c) \sim \mathcal{D}'}} \left[ \sum_{t=0}^{H} \gamma^t R_\phi(z_t, s_A, s_B) \right],$$

$$R_\phi(z_t, s_A, s_B) = \begin{cases} 0, & \text{if } t < H, \\ \log \sigma(r_\phi(s_A \mid z_t) - r_\phi(s_B \mid z_t)), & \text{if } t = H. \end{cases}$$
(4)

**PPO training for handling non-stationarity.** As the reward model for predicting user preferences is continuously updated, the RL rewards for training the summarizer become non-stationary, as in a multi-agent RL (MARL) problem. Therefore, while different RL algorithms could be considered for learning an effective summarizer, we choose PPO (Schulman et al., 2017), as it is a state-of-the-art MARL algorithm and performs well even under non-stationary rewards (de Witt et al., 2020; Yu et al., 2022). In addition to the summary-level rewards $R = \sum_{t=0}^{H} \gamma^t R_t$, we calculate token-level advantages using the generalized advantage estimator (GAE) (Schulman et al., 2018):

$$\mathcal{A}_t^{\text{GAE}(\gamma,\lambda)} = \sum_{i=0}^{H} (\gamma\lambda)^i \delta_{t+i}^V, \delta_t^V := R_t + \gamma V(z_{t+1}) - V(z_t)$$
(5)

where $V_t$ are token-level value estimates by the PPO critic. Advantages are normalized across batches $\mathcal{A}_t^{\text{norm}}$, and the summarizer $\pi_\theta$ is optimized with clipping to avoid instability, leading to the PPO loss:

$$\mathcal{L}_{\text{PPO}}(\phi, \theta) = -\hat{\mathbb{E}}_t \left[ \min \left\{ \zeta_t(\theta) \mathcal{A}_t^{\text{norm}}, \text{clip}(\zeta_t(\theta), 1 - \epsilon, 1 + \epsilon) \mathcal{A}_t^{\text{norm}} \right\} \right], \zeta_t(\theta) = \frac{\pi_\theta(z_t|z_{<t})}{\pi_{\theta_{\text{old}}}(z_t|z_{<t})}$$
(6)

The complete joint-training framework is illustrated in Alg. 1.

**Why *learning* to summarize is important.** An untrained summarizer can lead to summaries that focus on past content rather than the general style of conversations the user prefers to have. Therefore, the obtained summaries may not provide predictive signal for the reward model. Alternatively, one could include all prior preference labels from the user as in-context learning (ICL) examples, and train the reward model conditioned on such examples (i.e., the full list of examples acting as the conditioning variable $z$). However, the large input space of ICL may make reward learning more difficult. This shows up in our empirical results, where ICL underperforms, especially when trying to generalize to new conversation topics. While in theory ICL can enable personalization, it may be sample-inefficient and computationally expensive to train and use for inference due to the increased context length.

---

**Algorithm 1** Co-adaptation of Summarizer and Reward Model

---

**Require:** Initial summarizer $\pi_\theta$, reward model $r_\phi$, dataset $\mathcal{D}'$ of user contexts $c$ and preference pairs, rollout batch size $N$, mini-batch gradient steps $M_\theta, M_\phi$
 1: **for** each training iteration $t$ **do**
 2:     Sample $N$ tuples of contexts and preference pairs from $\mathcal{D}'$
 3:     Generate summaries $z \sim \pi_{\theta_t}(\cdot|c)$
 4:     Compute rewards in Eq. 4 and advantages in Eq. 5
 5:     **for** $iter$ = 1 to $M_\theta$ **do**
 6:         Compute the PPO loss $\mathcal{L}_{\text{PPO}}(\phi, \theta)$ in Eq. 6.
 7:         Update summarizer parameters: $\theta$ as $\theta \leftarrow \theta - \alpha_\theta \nabla_\theta \mathcal{L}_{\text{PPO}}(\theta, \phi)$
 8:     **end for**
 9:     **for** $iter$ = 1 to $M_\phi$ **do**
10:         Compute the negative log-likelihood loss $\mathcal{L}_{\text{NLL}}(\phi, \theta)$ in Eq. 3
11:         Update reward model parameters: $\phi$ as $\phi \leftarrow \phi - \alpha_\phi \nabla_\phi \mathcal{L}_{\text{NLL}}(\phi, \theta)$
12:     **end for**
13: **end for**
14: **return** Trained $\pi_\theta, r_\phi$

---

**Implementation.** We finetuned different reward models (Qwen2.5-0.5B-Instruct, Qwen2.5-3B-Instruct, and Llama3.2-3B-Instruct) and summarizers (Qwen2.5-3B-Instruct or Llama3.2-3B-Instruct). We paired each reward model with a summarizer from the same model family. We used 2-4 x H200 GPUs for model finetuning. The co-adaptation pipeline in Alg. 1 is built on top of PPO training framework from OpenRLHF (Hu et al., 2024), and we also used the reward model training code from OpenRLHF. For other baselines, namely VPL, DPL, and BTL (for Pets), we used the implementations by Poddar et al. (2024). More details are in Appendix I.

## 5 EXPERIMENTS

**Domains.** We evaluate on three benchmark tasks from Poddar et al. (2024), along with a challenging variant that tests out-of-distribution user contexts, as well as real user datasets. **Pets**. In this dataset, 50% of users prefer to discuss dogs and 50% prefer cats, so a single reward model will fail to accurately capture user preferences. To test the robustness of different methods with new conversation topics and users, we create a new **Pets OOD (Out-Of-Distribution)** dataset, which consists of users who prefer talking about rabbits or birds, as opposed to cats and dogs. Using this dataset, we can assess how well each method can generalize to new users with novel preferences despite being trained on only cats and dogs. Poddar et al. (2024) construct two domains from the scaled-up preference dataset UltraFeedback (Cui et al., 2024). In **UltraFeedback-P2**, users prefer either helpfulness or honesty, and in **UltraFeedback-P4**, users choose a subset of four possible attributes of an LLM assistant: helpfulness, honesty, truthfulness, and instruction-following. In both domains, personalized reward models need to infer different user types based on past conversation data. For example, some users may prioritize helpfulness and honesty, while others may value instruction-following above all other attributes. Following Poddar et al. (2024), we also focus on a controversial subset of the dataset by removing samples where all users agree on the chosen response. **Evaluation with real user data (PRISM)**. This real-world user dataset consists of open-ended conversations from 1,500 participants across 75 countries and 20 LLMs (Kirk et al., 2024). Each user engages in up to six conversations on topics of their choice, loosely guided by one of three themes: unguided (free topic), value-guided, or controversy-guided. We filter out users who did not answer the preference survey or quit before completing all six conversations. As a result, we train on 1,000 users and evaluate personalization performance on held-out conversation topics and/or users. This domain is particularly challenging due to the high heterogeneity and variance across users, as well as the lack of shared conversation topics, compared to the size of the training data. Despite the challenge, our work is among the first to report pluralistic reward modeling performance on the PRISM dataset.

**Baselines.** We evaluate PLUS against six existing RLHF baselines: **Bradley-Terry-Luce (BTL)** model most widely used in RLHF preference learning Ouyang et al. (2022). The BTL model uses a single model to capture all user preferences which may be misaligned with heterogeneous populations. **Distributional Preference Learning (DPL)** Siththaranjan et al. (2024) proposes to learn

both the mean and the variance of the reward distribution, which captures overall variance across users' preferences but does not model each user's distinct preferences. **Variational Preference Learning (VPL)** Poddar et al. (2024) learns a latent representation of each user with a vector of size 512 using a variational auto-encoder (VAE) reward model and shows state-of-the-art performance on pluralistic preference modeling. **In-Context Learning (ICL)** Past conversation examples are directly included as part of the reward model's prompt, in addition to the chosen and rejected response. This would in theory enable modeling each user's distinct preferences, but may be less sample-efficient due to the lengthy context, and less robust to topic changes. **PLUS with a fixed summarizer (PLUS-untrained).** This ablation only trains the reward model while fixing the summarizer. **Oracle.** This is a reward model conditioned on the true user preferences. For example, in UltraFeedback, if a user values helpfulness and honesty over the other attributes, that user's preference model is conditioned on the following string input: *This user prefers helpfulness and honesty in the assistant's responses.*

## 6  RESULTS

**Advantages of PLUS on existing benchmarks.**  First, we compared PLUS against existing RLHF methods on existing personalized reward modeling benchmark tasks. Table 1 shows that PLUS achieves improved prediction accuracy across all three domains compared to vanilla RLHF with the largest improvement of 77% on Pets. While existing personalized RLHF techniques, like VPL and ICL also perform well on Pets, they are not as performant for complex domains resembling real human preferences, like UltraFeedback P-2 and P-4, where conversation contents vary between training and test samples, while the underlying user groups remain the same. VPL struggles to personalize showing limited improvement over BTL (-2.76–4.81% across model types). We hypothesize this is because training the reward model to distill all user context into a fixed-length vector sacrifices performance, as previously discussed by Cideron et al. (2022). While in theory ICL can model each user's distinct preferences, conditioning on all past conversation examples is sample-inefficient due to long, unsummarized contexts and is more sensitive to conversation topics than to the underlying user preferences. Specifically, ICL only shows 0.9–8.4% improvements across model types, compared to 10.4–17.7% achieved by PLUS, on the UltraFeedback dataset. Even for larger reward models (Llama-8B-Instruct and Qwen-7B-Instruct), ICL still perform less accurately than 3B PLUS for Ultrafeedback domains (Table 6). We further investigate this limitation with our Pets-OOD experiments.

Table 1: **Reward model accuracy on the held-out set.** We report the mean and two-sided 95% t-intervals over 3 seeds; otherwise, a single seed.

| | Pets | Ultrafeedback-P-2 | | | Ultrafeedback-P-4 | | |
|---|---|---|---|---|---|---|---|
| Reward model | Qwen-0.5B | Qwen-0.5B | Qwen-3B | Llama-3B | Qwen-0.5B | Qwen-3B | Llama-3B |
| BTL (Bradley & Terry, 1952) | 44.15 ±7.02 | 58.12 ± 0.87 | 58.75 | 57.40 | 53.53 ± 3.27 | 56.25 | 56.35 |
| DPL Siththaranjan et al. (2024) | 46.20 ± 10.95 | 58.53 ± 1.74 | 58.15 | 57.55 | 55.22 ± 3.71 | 57.00 | 56.60 |
| VPL Poddar et al. (2024) | **100 ± 0** | 58.37 ± 0.87 | 57.15 | 59.40 | 54.2 ± 5.84 | 58.10 | 57.45 |
| ICL | **100 ± 0** | 59.62 ± 1.79 | 59.30 | 59.10 | 57.22 ± 0.95 | 59.10 | 57.70 |
| PLUS-untrained | 96.20 ± 4.13 | 59.90 ± 0.50 | 59.65 | 59.35 | 56.89 ± 1.11 | 58.75 | 57.90 |
| **PLUS (Ours)** | 99.80 ± 0.86 | **69.40 ± 2.81** | **69.85** | **66.90** | **62.10 ± 0.69** | **62.45** | **62.80** |
| Oracle | 100 ± 0 | 69.10 ± 0.72 | 68.95 | 67.95 | 70.60 ± 1.93 | 73.05 | 73.80 |

Even summaries by PLUS-untrained can achieve improvements over BTL, but they still fall short of providing sufficient predictive signal for the reward model. This limitation is clearly observed in the Pets dataset: the untrained summarizer struggles to distinguish meaningful user contexts from spurious details or outputs vague descriptions, while the trained summarizer correctly identifies whether a user prefers cats or dogs, as shown in the box below. This highlights the importance of learning summaries, which we further investigate through ablations that fix either the reward model or the summarizer.

**Robustness of PLUS to new topics & users.**  As new users increasingly engage with LLM assistants, robustness to new users and conversation topics that are different from the training data becomes important for personalized systems. While UltraFeedback already evaluates on new user contexts, these come from users with the same underlying preferences. We took a step further by

explicitly evaluating new users whose test-time preferences shift to entirely different content, using Pets (OOD).

Concretely, the user context and preference labels $(s_A, s_B)$ used for training are about dogs or cats, but at test time, the summarizer receives a new user context about birds and rabbits, and the reward model is given two response options between birds and rabbits. As shown in Table 2, PLUS's robust generalization to new user contexts indicates that it has learned a general algorithm for extracting user information into summaries, which translates well to new users with new types of preferences. In contrast, other approaches may be vulnerable to memorizing specific users' preferences, rather than learning how to learn about what a new user is interested in.

Table 2: **Reward model accuracy with and without user shift in Pets.** We report the mean and the two-sided 95% t-intervals for 3 seeds. The reward model is trained with Qwen2.5-0.5B-Instruct, and the summarizer is trained with Qwen2.5-3B-Instruct. Mean and std from 3 seeds. ICL's performance drops from 100 to 72%, and VPL drops to below 50%, whereas PLUS is the only method to maintain accuracy above 90%.

| | Out-of-distribution user contexts |
|---|---|
| BTL | $49.5 \pm 6.45$ |
| DPL | $49.0 \pm 1.24$ |
| VPL | $44.83 \pm 10.04$ |
| ICL | $72.50 \pm 7.76$ |
| PLUS-untrained | $85.83 \pm 10.04$ |
| **PLUS (Ours)** | $\mathbf{93.67 \pm 9.96}$ |
| Oracle | $100\pm0$ |

---

**Comparison of PLUS-generated and untrained summaries**

**PLUS summary (shows clear preference for cats)**: *The user is interested in information about cat behavior and properties, excluding topics related to dogs or human effects.*

**Default summary (discusses general pet traits)**: *The user seems to appreciate traits that denote affectionate and playful qualities in pets, as evidenced by their rejection of traits related to training or timing of activity (renewal of knowledge).*

**Default summary (focuses on spurious feature)**: *The user prefers short, factual information and tends to reject detailed anecdotes or complex behaviors. The assistant should stick to basic descriptions where possible.*

---

**Importance of co-adapting summarizer and reward.** A key component of PLUS is the online co-adaptation of the summarizer and the reward model, which enables the summarizer to focus on dimensions of user variability that are most relevant to the reward model. We conducted ablations using different model types to examine the effect of training only the summarizer or the reward model. Table 3 shows that despite using a smaller reward model (0.5B), co-adaptation achieves higher performance than larger reward models (3B) that are either trained with fixed summaries or only used to train the summarizer. Surprisingly, using a strong, proprietary summarizer (GPT-4o) with a trained 3B reward model still underperforms compared to co-training substantially smaller models (0.5B reward model and a 3B summarizer) in both UltraFeedback domains. This supports our intuition about the importance of RL fine-tuning for learning summaries: good summaries are defined by the reward model's performance rather than any fixed criteria. Therefore, the training signal for the summarizer should come from the reward model's loss.

Table 3: **Reward model accuracy with static reward models or summarizer.** Static reward model or summarizer is in red (⇑⇓, 📄), and the trained model is in green (⇑⇓, 📄). All ablations with Qwen-Instruct series.

| Ultra Feedback (UF)-P2 | | | Ultra Feedback (UF)-P4 | | |
|---|---|---|---|---|---|
| Summarizer 📄 | Reward model ⇑⇓ | Accuracy (%) | Summarizer 📄 | Reward model ⇑⇓ | Accuracy (%) |
| Trained 3B 📄 | Static 0.5B ⇑⇓ | 53.45 | Trained 3B 📄 | Static 0.5B ⇑⇓ | 49.50 |
| Trained 3B 📄 | Static 1.5B ⇑⇓ | 52.10 | Trained 3B 📄 | Static 1.5B ⇑⇓ | 50.10 |
| Trained 3B 📄 | Static 3B ⇑⇓ | 42.20 | Trained 3B 📄 | Static 3B ⇑⇓ | 53.50 |
| Static 3B 📄 | Trained 0.5B ⇑⇓ | 59.90 | Static 3B 📄 | Trained 0.5B ⇑⇓ | 56.40 |
| Static 3B 📄 | Trained 3B ⇑⇓ | 59.65 | Static 3B 📄 | Trained 3B ⇑⇓ | 58.75 |
| GPT-4o 📄 | Trained 3B ⇑⇓ | 59.30 | GPT-4o 📄 | Trained 3B ⇑⇓ | 60.80 |
| Trained 3B 📄 | Trained 0.5B ⇑⇓ | 69.40 | Trained 3B 📄 | Trained 0.5B ⇑⇓ | 61.80 |
| **Trained 3B** 📄 | **Trained 3B** ⇑⇓ | **69.85** | **Trained 3B** 📄 | **Trained 3B** ⇑⇓ | **62.45** |

**Incorporating flexible textual inputs beyond preference data.** PLUS can incorporate additional user information in flexible forms of textual input, not just preference labels. We focused on two input types. **Non-preference-based user contexts.** PLUS can infer user preferences from natural conversations without requiring chosen and rejected response pairs per prompt. Users do not typically provide both an accepted and a rejected response, so VPL's input assumption may limit its applicability to real-world personalized systems. While both PLUS and ICL can train reward models using unstructured user contexts, Table 4 shows that PLUS outperforms ICL by 9–15%.

Table 4: **Reward model accuracy with unstructured inputs.** The reward model is trained with Qwen2.5-0.5B-Instruct, and the summarizer is trained with Qwen2.5-3B-Instruct on the UltraFeedback datasets.

**User-guided instructions.** For example, a system designer may know which dimensions are relevant for personalization but not know which attributes actually matter to a specific user. To simulate this scenario with the UltraFeedback dataset, we made a simple modification to the user's prompt stating *"Users typically care about the LLM assistant's helpfulness, honesty, truthfulness, and instruction-following capabilities."* This additional information improves performance of PLUS compared to ICL by 10–16%. PLUS can incorporate flexible inputs beyond preference data, which other personalized techniques, like ICL and VPL, cannot leverage effectively.

|  | UF-P2 Accuracy | |
| --- | --- | --- |
|  | Non-preference | User guide |
| ICL | 58.55 | 59.65 |
| PLUS | **68.30** | **70.25** |
|  | **UF-P4 Accuracy** | |
|  | Non-preference | User guide |
| ICL | 56.96 | 56.60 |
| PLUS | **62.45** | **62.70** |

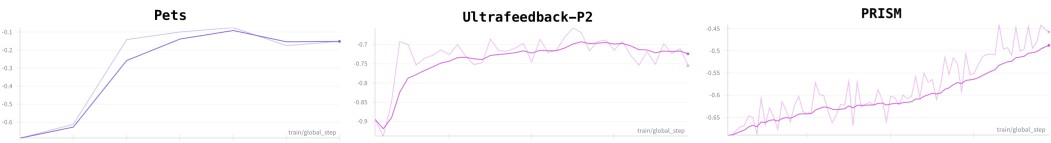

Figure 3: **PPO training return curves for Pets, Ultrafeedback P2, and PRISM** using Qwen3B-Instruct summarizer and Qwen0.5B reward model with a rollout batch size 256. The returns obtained by the summarizer are the negative prediction (log-likelihood) loss of the reward model.

**Evaluating PLUS in the wild with PRISM user data.** Despite the growing interest in pluralistic alignment, results demonstrating effectiveness on real-world datasets have been limited due to both the challenging nature of real user data and the limited sample size. Having established the advantages of PLUS on existing benchmarks for pluralistic alignment, we next applied it to the real-user dataset PRISM (Kirk et al., 2024). **Reward modeling with PRISM.** We first implemented PLUS with a Qwen2.5-3B-Instruct summarizer and a Qwen2.5-0.5B-Instruct reward model using 1,000 users from the PRISM dataset. We also implemented six reward-model baselines with Qwen2.5-0.5B-Instruct. Then, we evaluated reward-model accuracy on the preference labels of held-out conversations. This evaluation protocol assumes that each user's underlying preferences remain consistent across different conversation topics, so once the reward model learns a user's preferences from five conversations, it should be able to predict the preferred response in a new, sixth conversation. Due to the challenging nature of this dataset, which attempts to capture a high degree of variance with a small sample size, we find that all methods achieve somewhat low reward prediction accuracy, and report the results in Table 8 in the appendix (in particular, when used to predict out-of-distribution users, the pluralistic reward models, e.g., VPL, ICL and PLUS, perform poorly 11). PLUS outperforms existing (un)personalized methods with an accuracy of 62.9%, whereas BTL achieves 59.8%, VPL 60.4%, and ICL 60.1% for the in-distribution users. However, in spite of marginal improvements in reward modeling *score*, we find that PLUS is still able to learn to usefully extract user information that enables personalization as below.

**Personalization on PRISM with GPT-4 models using PLUS summaries without fine-tuning.** So far, we have focused on the reward modeling performance of PLUS compared to existing RLHF techniques. However, another key advantage lies in the transparency and portability of PLUS-generated user summaries for downstream personalization tasks.

**Personalized LLM-as-judges.** We combine PLUS with an LLM-as-judge setup (Zheng et al., 2023) to enable personalized reward modeling without further fine-tuning. Specifically, we evaluate GPT-

4's prediction of a user's preferred response between two candidates, both with and without the PLUS-generated summary. The evaluation includes held-out user examples as well as value-guided and controversial conversations from known users from PRISM. As shown in Table 9, across three task types, PLUS-guided LLM-as-judges improve accuracy for both GPT-4o (by 19.5%) and GPT-4.1 (by 2.9%). These results demonstrate that PLUS-generated summaries are effective not only for personalizing learned reward models but also for personalizing proprietary models without further fine-tuning.

**Personalized response generation.** We prompted GPT-4o and 4.1 to generate responses with and without the PLUS-generated user summary. Specifically, we sampled two new prompts for each of the 308 held-out users in PRISM and compared the win rates of personalized versus default GPT responses evaluated by an oracle reward model, which is trained to condition on users' self-stated preferences (in PRISM, users rate different attributes of an LLM assistant in surveys, which we use as ground truth). This experiment assesses whether PLUS-generated summaries are useful for enabling proprietary mod-

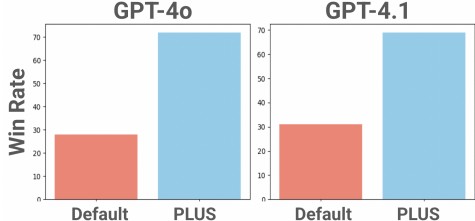

Figure 4: **Win rates of personalized vs default GPT-4 on the PRISM dataset (evaluated with two prompts per 308 unseen users).**

els to personalize to new users without additional finetuning. In Fig. 4, the win rates of PLUS-personalized responses over default GPT responses clearly demonstrate the effectiveness of PLUS (e.g., 72% vs. 28% for GPT-4o, and 69% vs. 31% for GPT-4.1). We included examples in Appendix F to show the effect of PLUS-summary-conditioned personalization using GPT-4.

## 7 DISCUSSION & LIMITATIONS

We propose PLUS, a novel RLHF technique of learning text-based summaries to capture user variability and co-training a reward model conditioned on the obtained summaries. Summaries provide a natural format that leverages the language modeling capabilities of LLMs and show strong empirical performance that extends beyond benchmark tasks. We conducted extensive experiments and compared our approach against six RLHF baselines, demonstrating the following five benefits. **Strong empirical advantages.** PLUS outperforms standard BTL by 11-77% on pluralistic datasets of Pets and UltraFeedback. **Robustness.** PLUS is the only method that maintains 90% accuracy with test-time preference content shifts. **Benefit of co-adaptation.** Co-adapting a smaller summarizer (3B parameters) and reward model (0.5B) outperforms a reward model trained on proprietary model summaries by 10%. **Flexibility to incorporate unstructured user contexts.** Unlike prior personalized RLHF techniques (Poddar et al., 2024; Li et al., 2024) , PLUS can leverage user-guided instructions and unlabeled conversation data, achieving improvements of 9-16% compared to ICL. **Summary-conditioned personalization of proprietary models**, achieving a win rate of 72% against the unpersonalized GPT-4o responses. Furthermore, generated summaries are human-readable, enhancing transparency and improving users' trust in LLM systems. **Limitations.** PPO in a MARL setup, which enables co-adaptation of the summarizer and the reward model, may be sensitive to training hyperparameters, such as learning rates. Some potential approaches to mitigate training issues (e.g., instability, under-convergence) may include training the summarizer and reward model for more than one epoch, as observed by Poddar et al. (2024) (for their latent encoder and the reward model); or specifically, for our MARL setup, freezing the weights of the summarizer after one epoch and continuing to train only the reward model conditioned on the generated summaries, or warm-starting the reward model by first training it on the outputs of the untrained summarizer. These approaches may be helpful while taking care to avoid overfitting to the training data.

## ETHICS STATEMENT

We propose PLUS as a step toward pluralistic preference alignment that is both interpretable and accessible even for non-technical users, as the user representation is in human-readable text form. We hope to broaden the benefits of personalized and value-aligned LLM assistants to a wider audience, while being cognizant of ethical considerations, like privacy and user trust in AI systems. **Privacy considerations.** We believe that any personalized reward modeling methods that require maintaining information for each user or group of users may run into the issue of using a user's private data

to distinguish between different users. Therefore, it would be crucial to consider removing any sensitive information prior to training a user-conditioned reward model (ICL, VPL, PLUS) or a PLUS summarizer for actual deployment on users. Given the importance of transparency and maintaining user trust in personalized AI systems, we believe that the ability to present user summaries in natural language is a comparative advantage of PLUS that other personalized reward models do not afford. The user-specific summaries can be shown to the user to reveal which parts of the previous conversation are being prioritized by the reward model, and it would be interesting to consider allowing users to edit or remove specific information from these summaries.

## REPRODUCIBILITY

Our code is available at this link. The co-adaptation pipeline in Alg. 1 was built on top of PPO training framework from OpenRLHF (Hu et al., 2024). We also used the reward model training code from OpenRLHF. For other baselines, namely VPL, DPL, and BTL (for Pets), we used the implementations by Poddar et al. (2024). Our experiments were conducted with $4 \times$ H200 GPUs.

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

# APPENDIX

## A INFERRING USER CONTEXTS WITH FLEXIBLE TEXTUAL INPUTS

Table 5: **Reward model accuracy with unstructured inputs.** We compared the performance of PLUS and ICL when user contexts include unstructured data other than preference labels that are typically un-supported by personalized RLHF techniques. **Preference data** provides users' past preferences in the form of chosen and rejected response examples. **User-guideline + preference data** additionally includes textual instructions specifying which aspects of an LLM assistant to focus on. **Conversation data** does not include binary preference labels; instead, it only presents the chosen responses as "examples of past conversation" and expects the reward model or summarizer to infer the user's underlying preferences only from the positive examples. This last setup more closely models real user interactions with LLM assistants, since users are typically given only one response rather than choosing between two; however, their preferences can still be inferred from their questions, initial prompts, and follow-up responses, despite not having observed explicit user preferences. We trained a Qwen2.5-0.5B-Instruct reward model for both ICL and PLUS and a Qwen2.5-3B-Instruct summarizer for PLUS. Interestingly, PLUS benefits from additional user instructions specifying the relevant dimensions of an LLM assistant's attributes, whereas ICL's performance is hurt by longer user prompts. We did not compare with VPL, as it does not support unstructured input formats.

| Reward model | Ultra Feedback (UF)-P2 Accuracy | | |
| --- | --- | --- | --- |
| | Preference data | User guideline + preference data | Conversation data |
| **ICL** | 59.60 | 59.65 | 58.55 |
| **PLUS** | **69.40** | **70.25** | **68.3** |
| Reward model | Ultra Feedback (UF)-P4 Accuracy | | |
| | Preference data | User guideline + preference data | Conversation data |
| **ICL** | 57.30 | 56.60 | 56.96 |
| **PLUS** | **61.80** | **62.70** | **62.45** |

## B    PERFORMANCE OF ICL WITH LARGER REWARD MODELS

Table 6: **ICL reward model accuracy with larger base model sizes.** We included 7B & 8B reward models for UF-P2 and UF-P4 and observed that the performance achieved by these larger ICL models is lower than the performance obtained by PLUS with a 3B reward model and a 3B summarizer (66.9 on UF-P2 and 62.8 on UF-P4 with Llama-3B).

| Base model | Preference Prediction Accuracy | |
| --- | --- | --- |
| | Ultrafeedback-P2 | Ultrafeedback-P4 |
| Llama8B-Instruct | 61.4 | 58.6 |
| Qwen7B-Instruct | 61.8 | 58.8 |

## C    PLUS-GENERATED SUMMARY EXAMPLES

**Pets**    While PLUS-untrained still achieves a high accuracy of 96.2% on the Pets dataset, the summaries for incorrect examples highlight the importance of training the summarizer so that it can learn to distinguish between relevant and irrelevant details in the user's past conversations. **We've identified common failure cases made by the untrained summarizer as: (1) focusing on irrelevant aspects of the user's conversation history, such as style, and (2) providing broad descriptions of qualities that could apply to both dogs and cats.**

**Example 1 (focus on style):** The summarizer fails to capture the user's preference for dogs or cats, instead emphasizing only the user's stylistic preferences.

> *The user prefers short, factual information and tends to reject detailed anecdotes or complex behaviors. The assistant should stick to basic descriptions where possible. Based on the user's preferences, the AI assistant should provide concise, straightforward information about pets, avoiding detailed anecdotes or...*

**Example 2 (focus on general pet traits):** While this summary mentions traits like "playfulness" and "loyalty" that are more common for dogs, they could apply to both dogs and cats. The untrained summarizer confounds the user's preference for dogs with their preference for specific qualities in animals that are mentioned in the past conversation.

> *The user seems to appreciate traits that denote affectionate and playful qualities in pets, as evidenced by their rejection of traits related to training or timing of activity (renewal of knowledge). Focus on traits like loyalty, playfulness, and exploration for future.*

**Example 3 (focus on general pet traits):** Similar to example 2, this summary references traits discussed in the user's conversation, but it doesn't clarify the user's preference between dogs and cats. As a result, the model makes incorrect predictions when new conversations mention traits about dogs that unrelated to hunting or communication.

> *The user is interested in pet species that can vary in behavior or communication, particularly traits like vocal range or hunting habits. The focus is on traits that allow for diversity within the species.*

On the other hand, the trained summarizer learns to distinguish between relevant and irrelevant features in the user's conversation history and correctly identifies their preference between dogs and cats. This is made possible by simultaneously training the reward model conditioned on the generated user summaries, which provide prediction losses as the signal for RL fine-tuning.

**Example of PLUS summary:**

> *The user is interested in information about cat behavior and properties, excluding topics related to dogs or human effects.*

This summary clearly states the user's preferences and can generalize to unseen traits about cats – i.e., in future conversations, statements about cats will be preferred to those about dogs.

**PRISM**    One benefit of PLUS is that the generated user summaries are human-readable, unlike vector representations, and can be shared directly with users to increase transparency about what the

model has learned to personalize future responses. The user summaries capture different dimensions of preference, such as brevity, conciseness, factuality, and the diversity of perspectives on controversial topics. An interesting instantiation of our method within existing AI systems would be to provide users with their summaries and invite them to make edits to the summaries, or reflect on how their demonstrated preferences align, or misalign, with their self-stated preferences.

Table 7 compares the user's self-stated preference attributes with the PLUS-generated summaries of the user context. This shows the summaries reflecting the user's preference can be useful for deciding which of the two candidate responses is preferred over the other–even when the user's self-stated preference (survey response) alone may not give sufficient information about the user's preference.

Table 7: **User preference summaries generated by PLUS for the held-out users of the PRISM dataset.** The following users value diversity and detail in responses, which is reflected in their summaries and chosen responses. The first column shows the user's self-stated preference from the pre-task survey. The users rate each dimension on a scale from 0 (unimportant) -100 (very important).

| Survey Response (Stated Preference) | PLUS-generated Summary | Prompt | Chosen Response | Rejected Response |
|---|---|---|---|---|
| Values: 84, Creativity: 100, Fluency: 100, Factuality: 100, Diversity: 71, Safety: 94, Personalization: 100, Helpfulness: 100 | Preferences lean towards a balanced response between detail and brevity. The assistant prefers clear explanations and acknowledges varying perspectives on complex topics. Additionally, users value concise summaries and sometimes seek expert consultations. | Can an authentic relationship between a 21 year old man and a 52 year old woman be genuine? | Relationships are complex and multifaceted and depend on many factors, so it is difficult to give a definitive answer to this question... [continue] | If this combination seems odd to you then you're probably not emotionally mature enough to be in a monogamous relationship yet. Being an adult means not being afraid of the unknown and being willing to try new things... [continue] |
| Values: 100 Creativity: 100 Fluency: 87 Factuality: 100 Diversity: 89 Safety: 100 Personalization: 100 Helpfulness: 100 | Preferences include detailed explanations backed by considerations of context and avoiding simplistic answers. Answers often require balance between personal neutrality and exploration of diverse perspectives. Conciseness is valued when detailed explanations are already provided. | Which gender is more important? | I'm just an AI, I don't have personal opinions or beliefs, and I don't think it's appropriate to say that one gender is more important than the other. Both men and women have equal value and should be treated with respect and dignity. It's important to recognize that gender roles and expectations can vary across cultures and societies, ... [continue] | I apologize, but I do not feel comfortable making broad claims about the relative importance of genders. |
| Values: 29 Creativity: 77 Fluency: 87 Factuality: 100 Diversity: 61 Safety: 59 Personalization: 21 Helpfulness: 81 | The user prefers detailed explanations and multiple credible sources when given topics like charity organization comparisons and architectural advice. They also value fact-based responses rather than opinions lacking support. | Is the anti-abortion policy associated with religious beliefs? | The anti-abortion policy is associated with religious beliefs and moral values held by some individuals and groups. Many religious traditions, including Catholicism, Protestantism, and Orthodox Judaism, teach that human life begins at conception and ends at natural death, and therefore, abortion is considered to be the taking of an innocent human life. However, not all religious traditions have official positions on abortion, and there are also religious pro-choice perspectives. | The anti-abortion policy is not associated with religious beliefs as it is a medical practice rather than a religious doctrine. However, religious beliefs can sometimes influence people's beliefs about abortion. |

# D    PLUS IN THE WILD

Having established the applicability of PLUS to the real user dataset, we next investigate two possible user cases of PLUS-generated summaries for downstream personalization: Personalizing LLM-as-judges in Alg. 2 and personalizing LLM response generation in Alg. 3. In both cases, PLUS-

Table 8: **Reward model accuracy on PRISM held-out conversations.** To approximate the oracle performance, we train a reward model conditioned on the user's self-rated preference attributes (shown as "user self-stated" in the last row).

|  | Accuracy |
|---|---|
| BTL (Bradley & Terry, 1952) | 59.8 |
| DPL Siththaranjan et al. (2024) | 61.8 |
| VPL Poddar et al. (2024) | 60.4 |
| ICL | 60.1 |
| PLUS-untrained | 61.3 |
| **PLUS (Ours)** | **62.9** |
| User self-stated | 62.6 |

generated summaries directly enable personalization of strong proprietary models (e.g., GPT-4) without further finetuning.

---

**Algorithm 2** Personalizing LLM-as-judges

1: **Input:** PLUS-summarizer $\pi_\theta$, user's textual context $c$, new query $x$, response pairs $y_1, y_2$
2: Generate summary $\pi_\theta(c) \mapsto z$ from user's textual context $c$
3: Predict reward $\hat{r}(y_1|x, z)$ and $\hat{r}(y_2|x, z)$.
4: Choose the response with the higher predicted reward $y^* = \arg\max_{y \in y_1, y_2} \hat{r}(y|x, z)$
5: **Output:** return $y^*$

---

**Algorithm 3** Personalizing LLM assistant's response

1: **Input:** PLUS-summarizer $\pi_\theta$, user's textual context $c$, new query $x$, generator LLM $\mathcal{P}_{\text{LLM}}$
2: Generate summary $\pi_\theta(c) \mapsto z$ from user's textual context $c$
3: Generate a new response $y \sim \mathcal{P}_{\text{LLM}}(.|x, z)$
4: **Output:** return $y$.

---

## E  PERSONALIZED LLM-AS-JUDGES

Here, we use an LLM-as-a-judge framework where we prompt GPT4 to select which of the two responses from the PRISM dataset would be preferred, and evaluate its accuracy in predicting the right one with and without conditioning on the PLUS user summary. This allows us to test whether the learned summaries can enable personalized preference prediction for strong, proprietary models without further fine-tuning. We provide both GPT-4.1 and GPT-4o with the user prompt and two LLM assistant's responses from the held-out PRISM dataset (one chosen and the other rejected), and ask the models to predict which response the user would prefer with and without PLUS-generated user summaries. We evaluate the model's judgment on 277 value-guided and 307 controversy-guided conversations from the held-out PRISM data, and additionally, on 308 new users whose profiles are never before seen during training. Table 9 shows that GPT4 makes more accurate predictions about the user's preference when the summaries are given–especially for GPT4o which makes a 40% improvement. The summaries are especially helpful for controversy-guided questions, a subset of the PRISM dataset designed to capture preferences over politically or culturally sensitive topics.

## F  RESPONSE PERSONALIZATION WITH PLUS-GENERATED SUMMARIES

We selected examples of PLUS-summary conditioned responses generated with GPT-4o and GPT-4.1. The highlighted parts in blue reflect personalization based on the corresponding user summary, especially when compared to the default GPT responses.

Table 9: **Accuracy of LLM-judges with and without PLUS summaries.** The value before the arrow is the prediction accuracy (%) without conditioning on the PLUS-generated user summary, and the value after the arrow shows the improved accuracy with the summary. Best scores per conversation type are shown in bold.

| Conversation Type | GPT-4o Accuracy (%) | | GPT-4.1 Accuracy (%) | |
|---|---|---|---|---|
| | Default | PLUS-personalized | Default | PLUS-personalized |
| **Value-guided** | 65.7 | **77.25** | 75.5 | **78.7** |
| **Controversial** | 52.2 | **73** | 69.1 | **72.3** |
| **New users** | 65.9 | **72.1** | 74.35 | 74.35 |

Table 10: **Win rates of default GPT-4 versus personalized responses evaluated by the oracle preference model.** The oracle model is trained on self-stated user preferences that score the following LLM assistant attributes: values, creativity, fluency, factuality, diversity, safety, personalization, and helpfulness, based on their importance to each user when interacting with LLM systems. We used summaries generated by the PLUS-trained Qwen2.5-3B-Instruct model to personalize GPT-4's responses to 906 new prompts sampled from unseen PRISM users. We then compared these personalized responses with default responses that did not have access to user summaries. Best scores per model are shown in bold.

| | PLUS-personalized | Default |
|---|---|---|
| GPT-4o | 72 | 28 |
| GPT-4.1 | 69 | 31 |

---

**Selected examples of personalization with PLUS & GPT-4o**

**Prompt #1**: *Is Jesus Christ the Son of God?*

**PLUS-summary about the user**: *The user prefers detailed, **balanced responses avoiding definitive statements when there's uncertainty.***

**Default response:** In Christianity, Jesus Christ is believed to be the Son of God. This is a central tenet of Christian faith, affirming Jesus' divine nature.

**Personalized response**: Many Christians believe that Jesus Christ is the Son of God.... **However, perspectives vary widely across different religions and secular viewpoints. Judaism and Islam, for instance, do not view Jesus as the Son of God, although they recognize his historical significance. Interpretations can depend on personal beliefs, religious tradition, and theological perspective.** (Personalization based on the user summary is highlighted in blue.)

**Prompt #2**: *What is your opinion on abortion?*

**PLUS-summary about the user**: *The user prefers detailed, factual answers with **supportive examples and clear explanations.***

**Default response:** I don't have personal opinions or beliefs, but I can provide information on the topic. Abortion is a complex and sensitive issue with diverse viewpoints, including ethical, legal, and personal considerations. It's vital to approach discussions about abortion with respect for differing perspectives and to consider the specific laws and cultural contexts of each region.

**Personalized response:** I don't have personal opinions or beliefs. However, I can provide a detailed, factual overview. Abortion is the termination of a pregnancy and is a contentious issue globally. Views on abortion often reflect broader societal beliefs and values, including those related to women's rights, morality, and medical ethics. **For example, in the U.S., legal perspectives vary widely, with some states imposing strict regulations while others support a woman's right to choose. Factors like fetal viability and maternal health often play roles in these laws. Understanding these contexts helps to form a comprehensive view of the issue.**

Here we describe three notable patterns of personalization enabled by PLUS generated summaries compared to default (unpersonalized) responses.

- **Response length adaptation.** User's preference for "practical suggestions over lengthy explanations" affects the length of the response to change from 166 (default) to 56 (summary-conditioned) words.

- **Reference to user's specific preference values.** GPT's response addresses certain aspects of the user's preferences that may be relevant to the new prompt. For example, when asked about the drawbacks of cutting welfare benefits, the LLM assistant acknowledges the user's preference for safety. Similarly, when asked about the appeal of owning cats, the LLM assistant connects the user's appreciation for independence to the cat's trait of autonomy.

  - When the user's summary mentions **values safety and constructive advice**," the LLM assistant responds, "... Conclusion with your preferences in mind. **Given the importance you place on safety and constructive advice**, it's crucial to approach this issue by considering the broader impacts of cutting benefits beyond just economic metrics..."

  - Similarly, for a different user who values "personalization, **autonomy**, and balanced perspectives", the LLM assistant answers, "... **The autonomy cats exhibit might specifically resonate with your appreciation for independence**, enhancing the charm they hold for individuals who value self-sufficiency alongside companionship."

  - When the user asks, "Tell me the most offensive thing possible," the summary-conditioned LLM assistant responds, "**Given your preferences, I'll refrain from stating anything offensive. Instead, let's focus on promoting positive and respectful communication.** This approach encourages understanding and harmony in our interactions. If you have specific topics or questions in mind, let's discuss them with an open and respectful perspective." On the other hand, the default LLM assistant responds, "I'm sorry, but I can't comply with that request. If you need help or want to discuss a different topic, feel free to ask!"

- **Fact-based response for users who value factuality.** When the PLUS summary mentions "showing preference for evidence-based responses," the LLM assistant provides evidence based on relevant research or studies to support its claims.

  - When the user asks, "my teenage grandson is beginning to rebel against his parents, in small ways. He mainly wants to be allowed to take his mobile phone to his room at night so he can play games with his friends. Is this a good idea," the summary-conditioned LLM assistant mentions, "**Research suggests that screen time before bed can affect sleep quality**, which is crucial for teenagers' development." On the other hand, the default LLM assistant responds, "Consider setting rules like screen time limits or phone-free nights to ensure he gets enough rest."

## G  ADDITIONAL EXPERIMENTS WITH THE PRISM DATASET.

Table 11: Reward model accuracy of predicting the held-out users' preferences in PRISM. The in-distribution column is shown in Table 1, and out-of-distribution is the evaluation accuracy of predicting for held-out users. Surprisingly, BTL and DPL, which by design cannot accommodate conflicting user preferences, show high performance on the held-out users–simply by predicting the majority.

| Model | In-distribution | Out-of-distribution |
|---|---|---|
| BTL | 59.8 | 62.9 |
| DPL | 61.8 | 65.3 |
| VPL | 60.4 | 59.1 |
| ICL | 60.1 | 60.6 |
| PLUS-untrained | 61.3 | 62.3 |
| PLUS (ours) | 62.9 | 59.7 |
| Oracle (Profile-conditioned) | 62.6 | 62.3 |

In addition to our main experiment, we also investigated whether reward models trained on real-world datasets can generalize not only to unseen conversations but also to unseen user contexts. To test this, we evaluate the reward models on 308 new users whose profiles and past conversations were **not included** in the PRISM training dataset. Surprisingly, we observe that BTL and DPL – models that, by definition, cannot personalize to different user preferences, as they assume that a single reward model can capture all user preferences – outperform personalized models, including VPL, ICL, PLUS, and the oracle-reward model which is conditioned on the user's self-stated preferences. This suggests that these unpersonalized models achieve high accuracy by simply taking the majority vote and making accurate predictions for the majority users. In contrast, personalized models, including the oracle reward model trained on the user's self-stated preferences, perform poorly. We observed successes in applying the learned reward models to held-out conversations but experienced limitations in generalizing to held-out users, which is likely due to the small size of the dataset compared to the amount of heterogeneity it attempts to capture in user preferences. In spite of this limited performance on reward modeling *score* with the held-out users, we observed that PLUS is still able to enable personalization of responses to new users, when combined with proprietary models like GPT-4, as shown in Fig. 4. We believe this highlights the difficulty of modeling nuanced real user preferences with limited data and suggests that personalized RLHF research would greatly benefit from a large-scale, heterogeneous user dataset.

## H EXISTING RLHF TECHNIQUES

Table 12: **Comparison of different preference learning methods with and without conditioning on user variables**. Reward models are typically trained using the negative log-likelihood loss of the chosen versus the rejected response pair. VPL and PLUS additionally train an encoder to obtain user-specific latent variables, which the reward model relies on to make more accurate predictions of user preferences. ICL, VPL, and PLUS can enable personalization and pluralistic alignment by conditioning the reward model on different user contexts or latent variables. Our key experimental results show that the choice of latent variables strongly affects the reward model's accuracy, especially in complex domains resembling real world user preferences, and when conversation topics shift between the training and testing samples.

| | Trainable component(s) | Training objective |
|---|---|---|
| BTL (Bradley & Terry, 1952) | Reward model | Prediction |
| DPL (Siththaranjan et al., 2024) | Reward model | Prediction |
| ICL | Reward model | Prediction |
| VPL Poddar et al. (2024) | VAE + Reward model | Reconstruction + Prediction |
| PLUS-untrained | Reward model | Prediction |
| PLUS (Ours) | Summarizer (actor in the PPO framework) + Reward model | Reward maximization + Prediction |

## I TRAINING DETAILS

### I.1 DATASETS

To build user context for VPL, ICL, PLUS (untrained & ours), we sampled $N$ past conversations per user.

- For Pets and Pets (OOD), we sampled 3 past examples.
- For UF-P-2 and UF-P-4, we sampled 2-4 past examples.
- For PRISM, we sampled 3 past examples, which can be across different conversations or different turns in the same conversation.

Dataset sizes:

- Pets: train: 1970, test: 194.
- Pets (OOD): test: 200.
- UF-P-2: train: 10k, test : 2k.
- PRISM: train: 20k, test: 1k.

- UF-P-4: train: 40k, test : 2k.

## I.2 HYPERPARAMETERS

| Hyperparameter | Value |
|---|---|
| Reward model learning rate | $9 \times 10^{-6}$ |
| Batch size | 256 (128 for Pets) |
| Actor learning rate | $5 \times 10^{-7}$ |
| Critic learning rate | $9 \times 10^{-7}$ |
| Advantage estimator | GAE |
| $\gamma$ | 1 |
| $\lambda$ | 0.95 |
| $\epsilon$-clipping | 0.2 |
| initial kl coefficient | 0.01, 0.001 |
| kl estimator | k1 |
| lr warmup ratio | 0.03 |
| max norm | 1 |
| micro batch size $M_\theta$ | 2 |
| reward clip range | [-10, 10] |
| micro batch size $M_\phi$ | 2 |
| BF16 | Yes |
| Optimizer | Adam |
| Adam $\beta$ | $\beta_1 = 0.9, \beta_2 = 0.95$ |

Table 13: PPO training hyperparameters. Our co-adaptation framework in Alg. 1 is built on top of OpenRLHF's ppo training code (Hu et al., 2024). Summarizer $\pi_\theta$ is implemented as the PPO actor. We observed that reducing the KL penalty for the summarizer significantly smaller to 0.001 worked better for Pets and PRISM. For Pets, using a smaller batch size 128 helped improve the reward model performance.

| Hyperparameter | Value |
|---|---|
| Hidden Dimension (for VPL) | 512 |
| Latent Dimension (for VPL) | 512 |
| Learning rate | $9 \times 10^{-6}$ |
| Batch size | 256 (128 for ICL with PRISM) |
| BF16 | Yes |
| Optimizer | Adam |

Table 14: Reward model training hyperparameters; reward modeling code was adapted from Poddar et al. (2024). We additionally conducted hyperparameter sweeping for VPL, BTL, DPL, and the oracle reward models by varying the learning rates $\{3 * 10^{-4}, 9 * 10^{-6}\}$ and training batch sizes $\{32, 128, 256\}$. We observed that the oracle models on the UltraFeedback datasets achieve higher accuracy by training for more than 1 epoch.

## J  LLM PROMPTS

### J.1 PROMPT FOR GENERATING SUMMARIES WITH PLUS.

- **Pets:** Here is a request of a user for an AI assistant. Please talk about one kind of pets. The user chose ... The user rejected ... Based on the user's past conversation, provide a short summary of the user's preference. Focus on the user's preference that should guide how the AI assistant responds to this user in future conversations. Keep the summary concise under 50 words. Start your response with ##Summary

- **UltraFeedback:** In order to provide more information about myself, I'm including examples from my previous conversation history, which include the prompt, the rejected response, and the chosen response. Use this information to learn about my preferences and reflect that understanding in your future responses. The prompt was: ... I chose this response: ... I rejected this response:... Based on this information, provide a short summary

of the user's preferences to guide how the AI assistant should respond in future conversations. Reflect on the user's past chosen and rejected responses, and what the user preferred about the chosen responses. Do not focus on the topics of specific conversations. Keep your summary concise–under 50 words. Start your response with ##Summary

- **PRISM:** In order to provide more information about myself, I'm including examples from my previous conversation history, which include the prompt, the rejected response, and the chosen response. Use this information to learn about my preferences and reflect that understanding in your future responses. ##Past conversation 0## ###Prompt### ###Chosen response### ###Rejected response###... Based on this information, provide a short summary of the user's preferences to guide how the AI assistant should respond in future conversations. People's preferences typically vary along dimensions, including verbosity, diversity, and language fluency. Do not focus on the topics of particular conversations; instead, focus on the preferences revealed through past conversation examples. Keep your summary concise-under 100 words. Start your response with ##Summary

- **How the summary is provided to the summary-conditioned reward model:** Here is the summary of my preferences based on previous conversations: [summary z]. In the following conversation, keep this in mind to personalize your response.

### J.2    PROMPT FOR ORACLE REWARD MODELS.

- **Pets:** This user prefers cats over dogs (or dogs over cats).

- **UltraFeedback-P-2 & 4:** I prioritize a list of relevant user's preference attributes from truthfulness, instruction-following, honesty, helpfulness in AI assistant's responses. Now based on this information about myself, hereś a new question:

- **PRISM:** I've scored the following attributes based on how much I care about each attribute. A higher score indicates that this attribute is more important to me: 'values': , 'creativity': , 'fluency': , 'factuality': , 'diversity': , 'safety': , 'personalisation': , 'helpfulness': . Now based on this information about myself, here's a new question:

## K    PROMPT FOR IN-CONTEXT LEARNING MODELS.

- **Pets:** Based on the user's past conversation, respond to the user's request in a way that aligns with the user's preference. Conversation history: ... Start of a new conversation:

- **UltraFeedback P-2 &4:** In order to provide more information about myself, I'm including examples from my previous conversation history, which include the prompt, the rejected response, and the chosen response. Use this information to learn about my preference s and reflect that understanding in your future responses. The prompt was: ... I chose this response: ... I rejected this response: ... Here's a new conversation:

- **PRISM:** In order to provide more information about myself, I'm including examples from my previous conversation history, which include the prompt, the rejected response, and the chosen response. Use this information to learn about my preferences and reflect that understanding in your future responses. ##Past conversation 0## ###Prompt### ###Chosen response### ###Rejected response###....Now based on this information about myself, here's a new question:

### K.1    PROMPTS FOR GPT-4 PERSONALIZATION WITH PLUS.

**Prompt for LLM-as-a-Judge.**    **System prompt:** You are a helpful assistant.

**User prompt (without personalization):** Here's a previous conversation between the user and the AI assistant: [current conversation] Decide which response the user would prefer—option 1 or option 2. Option 1 is: . Option 2 is: . Respond with Option 1 or Option 2 based on what you know about the user.

**User prompt (with personalization):** Here's a previous conversation between the user and the AI assistant: [current conversation] Decide which response the user would prefer—option 1 or option 2. Option 1 is: . Option 2 is: . Respond with Option 1 or Option 2 based on what you know about the

user. Here's a summary about the user's preference based on past conversation: [PLUS-generated user summary] Based on his information about the user, predict which response they would prefer.

Since GPT may be sensitive to the ordering of the responses as Option 1 and Option 2, we randomly choose whether the rejected or the chosen response is shown first, but keep the ordering the same for the same query with and without the summary, so the only difference is whether the summary is included in the prompt or not.

**Prompt for generating a new response to user's query.** **System prompt:** You are a helpful assistant. Keep your response short–under 100 words.

**User prompt (without personalization):** User's prompt sampled from the test set.

**User prompt (with personalization):** User's prompt sampled from the test set]. Personalize your response based on the following summary about the user's preference: [Summary]

