# OpenReview forum: "Learning to summarize user information for personalized reinforcement learning from human feedback"
_ICLR.cc/2026/Conference — ICLR 2026 Poster_

### Official Review · Reviewer_Kjwp · 2025-10-27

**Soundness:** 3
**Presentation:** 3
**Contribution:** 3
**Rating:** 8
**Confidence:** 4

**Summary:**

This paper proposes an approach for learning a reward model that is conditioned on a summary of a user profile. This means that the reward model can be personaliized. The approach jointly learns a summarization model with the reward model to ensure that the optimal info is included in the summary. The paper reports results showing that by doing this, the reward model accuracy improves and has a higher win rate on the PRISM dataset

**Strengths:**

This paper provides strong results showing that PLUS can better personalize. The authors show results on the PRISM dataset. this is a diverse, multicultural dataset and reflects what an LLM may encounter in the real world. Furthermore, PLUS shows robustness to new topics and users which is more difficult for approaches that, for example, assign a new user an id rather than a language-based summary.

**Weaknesses:**

The authors could add some important details related to training such as convergence criteria. Furthermore, PPO in a MARL setup is highly sensitive to learning rate and other parameters. The authors should discuss the possibilty of instability in training in the limitations section.

It would be interesting if the authors provided quantitative results to back up their claim that the personal embeddings dont capture information as well as the language summary - perhaps via mutual information measures between the user info and the summary/embedding

The paper would benefit from providing confidence intervals for results since some of the improvements are small and only tested over 3 seeds

The results in Table 10 in the Appendix are noteworthy and should be discussed in the paper.

A human evaluation of the summaries would be interesting to include to determine if better summaries are actually produced by training in this way. Its possible that the reward model learns to exploit the summarizer in ways that dont actually produce better summaries.

**Questions:**

Why is the oracle accuracy so low at times?

Can the authors comment on the privacy considerations of this approach compared to others?

Could the authors add details and discussion on the actual training of the summarizer and reward model?

Could the authors include quantitative results showing how well their language-based summarization captures the user compared to embedding approach?

---

> ### Author Response · Authors · 2025-11-21
> **Thank you for your feedback!**
>
> # Possibility of instability in training #
> - Thank you for this suggestion, we included this point in the limitations and added the training curves for the PLUS summarizer in Fig. 3.
>
>
> # Report confidence intervals #
> - We followed your suggestions and changed the multi-seed results in Table 1 and Table 2 to show the mean and the two-sided 95\% t-intervals. We also added more runs to Ultrafeedback-P4 and report the mean and the two-sided 95\% t-intervals. We show the updated columns from Table 1 and 2 below:
>
> | Reward model | Pets (Qwen-0.5B) | Pets OOD (Qwen-0.5B) | Ultrafeedback-P-2 (Qwen-0.5B) | Ultrafeedback-P-4 (Qwen-0.5B) |
> |----------|----------|----------|----------|----------|
> | BTL | 44.15 $\pm$ 7.02 | 49.50 $\pm$ 6.45 | 58.12 $\pm$ 0.87  | 53.53 $\pm$ 3.27  |
> | DPL | 46.20 $\pm$ 10.95 | 49.00 $\pm$ 1.24 | 58.53 $\pm$ 1.74  | 55.22 $\pm$ 3.71 |
> | VPL | **100 $\pm$ 0**  | 44.83 $\pm$ 10.04 | 58.36 $\pm$ 0.87  | 54.20 $\pm$ 5.84 |
> | ICL | **100 $\pm$ 0** | 72.50 $\pm$ 7.76   | 59.62 $\pm$ 1.79 | 57.22 $\pm$ 0.95  |
> | PLUS (untrained summarizer) | 96.20 $\pm$ 4.13  | 85.83 $\pm$ 10.04  | 59.90 $\pm$ 0.50  | 56.89 $\pm$ 1.11  |
> | PLUS (Ours) | 99.80 $\pm$ 0.86  | **93.67 $\pm$ 9.96** | **69.40 $\pm$ 2.81** | **62.10 $\pm$ 0.69** |
> | Oracle | 100 $\pm$ 0 | 100 $\pm$ 0 | 69.10 $\pm$ 0.72 | 70.60 $\pm$ 1.93  |
>
> # Why is the oracle accuracy low at times? #
> - Generally low reward modeling accuracy has also been observed by the earlier RLHF work (by Ouyang et al., 2022), where they report the reward model’s accuracy for predicting human-preferred outputs to be 72.4 $\pm$ 0.4 and 69.6 $\pm$ 0.9 (please refer to Appendix E.2).
> - In our setting, the oracle reward model for PRISM is conditioned on the user's self-stated preferences (i.e., the user's profile), which specify how they would like a language-model assistant to respond. Since there is no ground-truth 'oracle' reward model for a real-user dataset like PRISM, we use this model conditioned on the user's self-statement as an approximation.
>
>
> # Privacy considerations of this approach compared to others #
> - Thank you for raising an important point! We added the following to our ethics statement. We believe that any personalized reward modeling methods that require maintaining information for each user or group of users may run into the issue of using a user's private data to distinguish between different users. Therefore, it would be crucial to consider removing any sensitive information prior to training a user-conditioned reward model (e.g., ICL, VPL, PLUS) or a PLUS summarizer for actual deployment on users. Given the importance of transparency and maintaining user trust in personalized AI systems, we believe that the ability to present user summaries in natural language is a comparative advantage of PLUS that other personalized reward models do not afford. The user-specific summaries can be shown to the user to reveal which parts of the previous conversation are being prioritized by the reward model, and it would be interesting to consider allowing users to edit or remove specific information from these summaries.
>
>
> # Comparisons of language-based summarization #
> - Thank you for suggesting interesting ideas for quantitative and qualitative evaluation, such as mutual information measures and human evaluation of the summaries. We would be excited to explore them in future work.
> ***
> Reference:
>
> Ouyang et al., (2022). Training language models to follow instructions with human feedback.

---

### Official Review · Reviewer_8x8P · 2025-10-31

**Soundness:** 3
**Presentation:** 3
**Contribution:** 3
**Rating:** 4
**Confidence:** 3

**Summary:**

This paper proposes the PLUS framework for modeling user-specific preferences. A text-generating summarizer produces a user summary $z$ from the user context $c$, and a reward model $r_\phi(· | z)$ is optimized conditioned on that summary to capture personalized preferences. The summarizer and the reward model are trained in an alternating fashion. Experiments on benchmarks such as Pets and UltraFeedback-P-2 demonstrate strong performance compared with baselines.

**Strengths:**

# strengths
1. Representing user preferences via natural-language summaries makes the user representation easy to understand and inherently interpretable, while avoiding issues with overly long raw contexts.
2. PLUS is evaluated across diverse datasets and settings (including OOD), providing evidence of its effectiveness.
3. PLUS achieves improvements over baselines in most settings and even surpasses strong baselines such as GPT-4o and GPT-4.1.

**Weaknesses:**

# weakness
1. Data sparsity per user. With only a few dialogs for each user (e.g., Pets uses 3, UltraFeedback uses 2–4, PRISM uses 3), can the reward model learn a reliable representation of that user’s preferences? Is the reward model potentially *under-converged* in such low-data regimes?
2. *Training stability.* Alternating optimization can be hard to converge. It would be important to provide *training curves for the reward model* to assess stability (e.g., accuracy,  variance across steps).
3. Cost and scalability. Building a reward model at the *user granularity* raises the question: when new users arrive, does the system need to update the reward model each time? If so, this could lead to *high computational cost*.
4. *Context-dependent preferences.* A single user’s preferences may shift across contexts, which questions whether modeling a single, static per-user profile is appropriate. For instance, a user may like cats in a general conversation about stray cats, yet under a context emphasizing ecological harm, the same user’s preference toward stray cats could diminish.

If the authors can provide additional results to address these concerns, I would be happy to raise my score.

**Questions:**

See above weaknesses

---

> ### Author Response · Authors · 2025-11-21
> **New results with context-dependent preferences & training curves in Fig. 3**
>
> # New results for context-dependent preferences #
> - Thank you for suggesting an interesting test case where a user generally prefers cats, but “under a context emphasizing ecological harm, the same user’s preference toward stray cats could diminish.” Based on this idea, we created a new version of Pets, which we call Pets (OOD context). In this experiment, the training set includes only cat- and dog-lovers but at test time cat-lovers who are also environmentally conscious show diminished preferences towards cats in the context of stray cats harming the environment and dogs helping preserve the environment (and similarly for environmentally conscious dog-lovers).
>
> - We evaluate ICL and PLUS with different reward model types on the accuracy of predicting the user’s context-dependent preferences across 200 samples.
>
> | Llama-Instruct Series | Reward model accuracy (eval n =200) |
> |----------|----------|
> | ICL Llama-3.2-1B |  0 |
> |  ICL Llama-3.2-3B |  22        |
> |  ICL Llama-3.1-8B |  0    |
> |  **PLUS Llama-3.2-3B**  | **83.5**       |
>
> | Qwen-Instruct Series | Reward model accuracy (eval n =200) |
> |----------|----------|
> | ICL Qwen2.5-0.5B |  32 |
> |  ICL Qwen2.5-1.5B |  0       |
> |  ICL Qwen2.5-3B |  12    |
> |  ICL Qwen2.5-7B |  30    |
> |  **PLUS Qwen2.5-3B**  | **99**  |
> - Even though the ICL reward models achieve 100% accuracy on the training set, they fail when new user contexts are added and their preferences shift as a result of these contexts. Even the bigger ICL models are outperformed by the 3B PLUS model. Surprisingly, we also find that larger reward models for ICL do not necessarily improve generalization performance to new contexts compared to smaller reward models (e.g., Llama-3B achieves 22% accuracy while the 8B model achieves 0).
> - On the other hand, PLUS can adapt to include information about the user’s new preference related to the environment along with their original pet preference. Here are some test-time generated summaries:
>
> E.g., Qwen3B-PLUS: ``The user prioritizes choices that benefit the environment and has preferences towards pet cat traits, suggesting eco-friendly and less resource-intensive living habits and cat behaviors. In the following conversation, keep this in mind to personalize your response.``
>
> Llama3B-PLUS: ``The user prioritizes environmental and social concerns, choosing options that promote sustainability, climate justice, and responsible tourism. They also value low-maintenance pets, like cats, and acknowledge their importance. The user rejects options that promote disposable materialism, emphasis on quick economic benefits, and excessive routine changes.``
>
> - These examples further illustrate that PLUS doesn’t learn static user summaries, but rather it trains a summarizer that can be deployed at test-time to accommodate users' context-dependent preferences.
>
> ***
> More details about this experiment setup: We used GPT-4o to generate (a) responses to the following question: “Do you prioritize environmental concerns over other needs when making lifestyle choices?” (100 sentences for prioritizing the environment over personal benefits and convenience), (b) 100 sentences about the ecological harm of stray cats / dogs, and (c) 100 sentences about the ecological benefits of pet cats / dogs. We sampled 3 past dialogs about pets, 3 about the environment to augment with the additional context: ```I've learned some new pieces of information about pets recently: {benefit of pet dogs sample 1} {ecological harm of stray cats} {benefit of pet dogs sample 2} Which pet should I get?``` Cat lovers should now prefer dogs, vice versa.
>
> ***
> # Cost and scalability #
> Regarding the reviewer’s question “when new users arrive, does the system need to update the reward model each time?,” we’d like to clarify that the **PLUS summarizer and the reward model do not need to be updated for every new user.** This is because the summarizer is trained on the data from many users and learns ***how to summarize*** them in a way that helps improve reward modeling accuracy. We show generalization with Pets OOD and also with Pets OOD contexts following the reviewer's suggestion.
> ***
> # Data sparsity per user #
> The goal of PLUS is not to output a static representation for each user in the training dataset, but rather to learn a way to take the information that is available about users and summarize it in a way that improves the reward model accuracy. It would be interesting to explore more complex real-life examples where more dialogs are available per user and can be incorporated as additional context to enrich the summaries, which could further improve reward modeling performance.
> ***
> # Provide training curves for PLUS #
> Thank you for the suggestions! We included the training curves of the PLUS summarizer for Pets, Ulrafeedback-P2, and PRISM in our revision as Fig 3. Please note that the returns obtained by the summarizer are the negative prediction losses of the reward model.

---

> > ### Comment · Reviewer_8x8P · 2025-11-27
> >
> > Thank you for the response. The additional results and the updated manuscript have resolved some of my concerns, so I have decided to raise my score. However, the updated training curves also seem to confirm my previously mentioned Weakness 1: in Figure 3, all the curves are still increasing, especially PRISM, and pets dataset only train 7 steps, which indicates that the model is under-converged?

---

> ### Author Response · Authors · 2025-11-27
>
> Thank you for your review and feedback! We agree that balancing training convergence and test-time performance is important. We included your point in our limitations section: "Some potential approaches to mitigate training issues (e.g., instability, under-convergence) may include training the summarizer and reward model for more than one epoch, as observed by Poddar et al., 2024; or specifically, for our MARL setup, freezing the weights of the summarizer after one epoch and continuing to train only the reward model conditioned on the generated summaries, or warm-starting the reward model by first training it on the outputs of the untrained summarizer. These approaches may be helpful while taking care to avoid overfitting to the training data."
>
> Please note that for PRISM, due to the high heterogeneity of conversation topics across users (i.e., there are no common conversation topics shared across users, since users are instructed to interact freely with the AI assistant and only asked to indicate the type of conversation they'd like to have: unguided, values-guided, or controversy-guided (see Appendix Q in Kirk et al., 2024)) we prioritized avoiding overfitting, so the model can generalize to novel conversations/users. With that said, we would be very excited to apply our approach in the future to scaled-up, pluralistic user datasets, where users have more dialogs available and have a mix of shared and diverse conversation topics.
>
> Reference:
>
> Poddar et al., 2024. Personalizing reinforcement learning from human feedback with variational preference learning.
>
> Kirk et al., 2024. The PRISM Alignment Dataset.

---

### Official Review · Reviewer_x1Wn · 2025-11-01

**Soundness:** 3
**Presentation:** 3
**Contribution:** 3
**Rating:** 6
**Confidence:** 3

**Summary:**

The submission proposes a method to learn personalized reward models called PLUS.
It consists of a summarization model, the output of which is used to condition a reward model.
The summarization model and reward model are trained in an iterative fashion, alternating between doing PPO updates on the summarization model and supervised crossentropy updates on the reward model conditioned on the generated summaries.
The user summaries are textual.
Experiments show that the proposed method leads to a notably higher reward modeling accuracy, using the resulting reward model for preference-tuning is not evaluated.

**Strengths:**

* Adapting to a user's individual preference is an important problem
 * The method is well motivated and very reasonable
 * The results show a notable improvement in RM accuracy

**Weaknesses:**

* The evaluation is limited to reward modeling, it was not tested whether the trained reward model can be used in a subsequent step to train a better policy. Recent research [1,2] has shown that a higher RM accuracy does not necessarily lead to a better performance after RL training. This is my main issue with this paper.

The abstract also claims a "25% improvement over the best personalized RLHF technique". This is incorrect, at most the paper can claim such an improvement over the "best personalized RM technique", not over the best RLHF technique.

* Experiments were only conducted with rather small models (<3B). Particularly for the ICL baseline, ICL is known to perform better for larger models [3]. It is thus unclear the proposed method is advantageous to ICL for larger reward models which would be used in practice.

 * Writing is unclear or incorrect at times, examples below in the minor issues

Minor issues:
 * "BTL reward model [...] is incapable of creating systems that can adapt to users' unique needs and and preference" the paper's own ICL results show the opposite. They are worse than the proposed method, but not fully incapable of adapting to the preferences.
 * GPT4 is not a state-of-the-art model
 * Figure 1: using a religious example seems prone to invite discontent, a non-religious, non-political example may be more inclusive
 * "zero-shot" is used inconsistently and incorrectly throughout the paper. In L295 it refers to generalization to the test-set, in L342 it refers to not training the summary model, which is not zero-shot and in L445 it refers to prompting GPT4 without fine-tuning it. None of these are "zero-shot" in the classical sense of generalizing to entirely unseen classes.
 *  L201: The proposed method is described as "two-stage sequential". To me this sounds like we first train the summarizer, then the reward model, and then we are done. Instead the method alternates between both steps. L210 "Once the summarizer is fixed" also sounds like we fix it only "once", contradicting the alternating training scheme. As it is key to the paper I would recommend revising the presentation.



References:

[1] Razin et al. "What Makes a Reward Model a Good Teacher? An Optimization Perspective", NeurIPS 2025

[2] Chen et al. "The Accuracy Paradox in RLHF: When Better Reward Models Don't Yield Better Language Models", EMNLP 2024

[3] Brown et al. "Language Models are Few-Shot Learners", NeurIPS 2020

**Questions:**

* Why were no RL experiments conducted with the trained reward model? It would significantly strengthen the paper.
 *

---

> ### Author Response · Authors · 2025-11-21
> **Added new results with larger ICL reward models & preference tuning.**
>
> # Bigger models for ICL #
> - Thank you for your feedback! We included 7 & 8B models for UF-P2 and UF-P4 and observed that they achieve lower preference prediction accuracy than PLUS implemented with a 3B model. (**PLUS achieves 66.9 on UF-P2 and 62.8 on UF-P4 with Llama-3B-Instruct**).
> | Model             | UF-P2 | UF-P4 |
> |-------------------|-------|-------|
> | Llama-8B-Instruct | 61.4  |   58.6    |
> | Qwen-7B-Instruct  | 61.8  |    58.8   |
> - We also added an ablation of Pets for testing generalization to preference shifts based on new contexts (please review [8X8P] for more details on this setup) and **similarly observe that the larger reward models for ICL do not necessarily improve prediction.**
>
> | Instruct Models | Reward model accuracy (eval n =200) |
> |----------|----------|
> |  ICL Llama-3.1-8B |  0    |
> |  **PLUS Llama-3.2-3B**  | **83.5**       |
> |  ICL Qwen2.5-7B |  30    |
> |  **PLUS Qwen2.5-3B**  | **99**  |
>
> - Even though the ICL reward models achieve 100% accuracy on the training set of cat and dog lovers, they fail when new user contexts are added (e.g., users prioritize environment over personal benefits), and particularly when the user’s preferences shift as a result (e.g., in the context of stray cats harming the environment, cat-lovers who are also environmentally conscious show diminished preferences for cats).
>
> [8X8P]: https://openreview.net/forum?id=Ar078WR3um&noteId=VHaxMHmNzt
> ***
> # Preference fine-tuning #
> - While we did not initially include results performing RLHF fine-tuning with our reward models because this was not performed in prior reward modeling papers [1, 2, 3], based on your suggestion we have decided to conduct these experiments.
> - We ***fine-tuned Llama1B- and 3B-Instruct models using 1000 training prompts and evaluated the percentage of responses that correctly mention the target pet*** across 100 test points (e.g., for cat-lovers, responses should include cats). We included the mean and the standard deviation from 3 seeds for Pets (cat- and dog-lovers), where we expect the performance to be similar for ICL and PLUS based on the reward modeling accuracy.
> | Reward model used for training | 1B on Pets | 3B on Pets | 1B on Pets OOD | 3B on Pets OOD |
> |----------|----------|----------|----------|----------|
> | BTL |  42 | 57 | 23 | 35
> | ICL | 85.33 ($\pm$ 4.16)  | **99.33 ($\pm$ 1.15)**  | 23 | 33   |
> | PLUS |  **95 ($\pm$ 2.65)** | **99.33 ($\pm$ 0.58)** | **70** | **86** |
>
> - Please note that all the reward models (BTL, ICL, and PLUS) are trained from Llama3B-Instruct using the same hyperparameter set. (We additionally tried a smaller KL term for ICL, 1e-2, 1e-3). For fair comparisons, all generator policies are given the same amount of user information (i.e., user’s past conversation + prompt: “Tell me about one kind of pet”).
>
> ## Why is ICL worse than PLUS for preference fine-tuning ##
> - We further investigate these results and find that they can be explained by the reward gap. We sampled 100 contexts for each cat- and dog-lover and checked the reward model's average reward for different animals. We computed the difference between the reward of the optimal arm and that of the next highest suboptimal arm divided by the max reward. The larger the reward gap, the easier it is to learn an optimal generator policy.
>
> **For cat-lovers: The reward gap (divided by max reward) for PLUS was 1.30 but for ICL was 0.33.**
> |  | **cats** | dogs | elephants | horses | birds | turtles | rabbits|
> |----------|----------|----------|----------|----------|----------|----------|----------|
> | PLUS |  0.252 | -0.592 | -0.470 | -0.507 | -0.296 | -0.075 | -0.295 |
> | ICL | 0.030 | 0.010  | 0.017| 0.011 | 0.010 | 0.010 | 0.020|
>
> **For dog-lovers: The reward gap (divided by max reward) for PLUS was 1.16 but for ICL was 0.33.**
> |  | cats | **dogs** | elephants | horses | birds | turtles | rabbits|
> |----------|----------|----------|----------|----------|----------|----------|----------|
> | PLUS |  -0.598 | 0.293 | -0.100 | -0.046 | -0.428 | -0.423 | -0.412 |
> | ICL | 0.012 | 0.027  | 0.015 | 0.011 | 0.011 | 0.008 | 0.018 |
> - We suspect that the optimal reward gaps between ICL and PLUS may affect the performance of the downstream preference fine-tuned policies (as observed with Llama-1B-Instruct models).
> ***
> - Corrections of minor issues: We incorporated them into our revision (e.g., revised Fig. 1 to show a non-religious example and added a content disclaimer, replaced “state-of-the-art” with “strong proprietary” for GPT-4, and made corrections about “zero-shot” and “two-stage sequential” and "best personalized reward model" in the abstract).
> ***
> References:
>
> [1] Siththaranjan et al. (2023). Distributional Preference Learning: Understanding and Accounting for Hidden Context in RLHF.
>
> [2] Zhao et al. (2023). Group Preference Optimization: Few-Shot Alignment of Large Language Models.
>
> [3] Poddar et al. (2024). Personalizing reinforcement learning from human feedback with variational preference learning.

---

> ### Comment · Reviewer_x1Wn · 2025-11-25
>
> Thank you for providing the additional experiments and for the rebuttal.
>
> The additional experiments are encouraging and seem to support the proposed method.
>
> I think this paper presents a well motivated approach, albeit for a relatively specific problem setting, and should be accepted.

---

> > ### Author Response · Authors · 2025-11-27
> >
> > Thank you again for your time to review and give feedback on our work!

---

### Meta-Review · Area_Chair_KbVT · 2026-01-06

**Summary:**

This paper proposes PLUS, a principled framework for personalized reinforcement learning from human feedback that conditions reward models on natural-language summaries of user preferences. By jointly training a user summarization model and a reward model in an online co-adaptation loop, the approach provides an interpretable and effective mechanism for pluralistic preference alignment. The paper is technically sound, well-motivated, and supported by extensive empirical results across multiple benchmarks, including out-of-distribution settings. While some training instability and under-convergence are observed, these issues are acknowledged as limitations and do not detract from the overall contribution or the validity of the results. Overall, the paper makes a strong and positive contribution to personalized RLHF.

**Reviewer Concerns:**

The major concerns raised by the reviewers included: limited evaluation on downstream reward-model-based RL, small-scale experiments, training stability and convergence, cost and scalability of user-conditioned reward models, and handling of context-dependent preferences. Through the rebuttal and revision, the authors provided substantial additional experimental results addressing reward modeling, downstream preference fine-tuning, generalization to new users and contexts, scalability considerations, and explicit evaluations of context-dependent preferences. These additions resolved the majority of concerns. The issue of training stability and possible under-convergence remains, as evidenced by increasing training curves in some datasets, but this limitation is clearly discussed by the authors and is relatively minor in scope.

**Reviewer Scores:**

Most concerns raised by Reviewer 8x8P were well addressed through added experiments, training curves, and expanded discussion, leading the reviewer to raise their score. Given the resolution of the key methodological and empirical concerns, it is reasonable to expect that this reviewer would increase their score from the initial rating of 4. Other reviewers’ concerns were similarly addressed, and all reviewers would likely retain their current scores following full discussion.

---

### Decision · Program_Chairs · 2026-01-26

Accept (Poster)